# Pyrogallol-Phloroglucinol-6,6-Bieckol Alleviates Obesity and Systemic Inflammation in a Mouse Model by Reducing Expression of RAGE and RAGE Ligands

**DOI:** 10.3390/md17110612

**Published:** 2019-10-28

**Authors:** Junwon Choi, Seyeon Oh, Myeongjoo Son, Kyunghee Byun

**Affiliations:** 1Department of Anatomy & Cell Biology, Gachon University College of Medicine, Incheon 21936, Korea; choijw88@gc.gachon.ac.kr (J.C.); mjson@gachon.ac.kr (M.S.); 2Functional Cellular Networks Laboratory, College of Medicine, Department of Medicine, Graduate School and Lee Gil Ya Cancer and Diabetes Institute, Gachon University, Incheon 21999, Korea; seyeon8965@gachon.ac.kr

**Keywords:** obesity, hypertrophy, *Ecklonia cava*, phlorotannin, RAGE, inflammation

## Abstract

*Ecklonia cava* (*E. cava*) can alleviate diet-induced obesity in animal models, and phlorotannins contained in *E. cava* help prevent hypertrophy-induced adipocyte differentiation. Receptor for advanced glycation end-products (RAGE) is well known to induce hypertrophy of visceral fat and to trigger inflammation substantially. While the relationship between RAGE and obesity and inflammation has been well-characterized, few studies describe the effects of phlorotannin on RAGE. In this study, we investigated the anti-obesity effects of pyrogallol-phloroglucinol-6,6-bieckol (PPB)—a single compound from the ethanoic extract of *E. cava*—mediated by a reduction in the inflammation caused by RAGE and RAGE ligands. In visceral fat, PPB (i) significantly inhibited RAGE ligands, (ii) reduced the expression of RAGE, and (iii) reduced the binding ratio between RAGE and RAGE ligands. Under lower expression of RAGE, RAGE ligands and their cognate binding, the differentiation of macrophages found in visceral fat into M1-type—the pro-inflammatory form of this immune cell—was reduced. As the M1-type macrophage decreased, pro-inflammatory cytokines, which cause obesity, decreased in visceral fat. The results of this study highlight the anti-obesity effects of PPB, with the effects mediated by reductions in RAGE, RAGE ligands, and inflammation.

## 1. Introduction

Obesity is a common phenomenon in modern society due, at least in part, to over-consumption and nutrient imbalance [1]. Obesity is a major cause of various diseases (e.g., cardiovascular diseases, metabolic disease, dementia) [2,3,4]. Increase of body weight can be observed/modeled in diet-induced obesity (DIO) animal models [5]. Notably, an increase of visceral fat and noticeable obesity are caused by excessive calorie accumulation in adipocytes as a form of triglycerides. DIO animal models also result in high levels of total cholesterol and triglycerides in the serum.

Hypertrophic conditions generate advanced glycation end-products (AGEs) and other receptor for advanced glycation end-products (RAGE) ligands (e.g., high-mobility group box 1 (HMGB1), S100β) capable of binding to a RAGE [6]. In mouse models of obesity, RAGE plays a role in inducing insulin and leptin resistance. On the other hand, there are some reports that RAGE knock-out mice had reductions in visceral fat hypertrophy; in RAGE-overexpressing mice, increased hypertrophy was reported [7,8]. Collectively, research confirms that RAGE plays a critical role in adipocyte hypertrophy [9]. Compared with lean adipocytes, hypertrophic adipocytes had higher expression of RAGE and RAGE ligands (e.g., AGE, HMGB1, S100β, free fatty acids (FFAs)) and inflammatory cytokines (e.g., (TNF-α), interleukin (IL)-1β) [10]. RAGE ligands expressed from hypertrophic adipocytes induce the infiltration of macrophages into fat tissue and differentiate into pro-inflammatory macrophages (i.e., M1-type) [11]. FFAs also induce macrophages to differentiate into M1-type by activating toll-like receptors (TLRs). M1-type macrophages inhabiting fat tissue showed an increase in RAGE expression and secretion of RAGE ligands [12,13]. 

RAGE is well known to serve as a multi-receptor, which induces immune-inflammatory reactions by activating macrophages to secrete inflammatory cytokines. For example, RAGE-RAGE ligand interactions lead to the activation of mitogen-activated protein kinases (MAP kinases) pathways that transduce inflammatory signals by phosphorylating p38, ERK1/2 and lead to an infiltration of immune cells [14]. M1-type macrophages increase the secretion of inflammatory cytokines and induce inflammatory responses in tissues by (i) inducing infiltration of immune cells (e.g., T-cells, B-cells, macrophage) and (ii) subsequent pathogenesis of autoimmune diabetes by activating CD4+ and CD8+ T lymphocytes and B lymphocyte [15]. On the other hand, the number of M2-type macrophages—the anti-inflammatory type—are decreased in obese individuals compared with lean individuals. In the lean state, M2-type macrophages are the dominant type of macrophage in visceral fat and serve to suppress inflammation. M2-type macrophages secrete anti-inflammatory cytokines (e.g., IL-10, arginase-1 (Arg-1)), which inhibits the differentiation of other visceral fat-inhabiting macrophages into the M1-type [16,17].

Currently, there are medications to treat obesity, which act by suppressing appetite or the absorption of fat from the intestine. However, side effects are reported for these treatments (e.g., headache, diarrhea, constipation, insomnia) [18,19]. Recently, naturally occurring phlorotannin—a component of cell walls in brown algae—has been increasingly investigated for its potential ability to prevent and reduce metabolic diseases by affecting fat metabolisms, such as anti-oxidant, anti-inflammatory, and anti-obesity properties [20]. Phlorotannin is abundant in brown algae, especially *Ecklonia cava*. *E. cava* is commonly found near the coast of Jeju island and is known to be a particularly rich source of phlorotannin [21]. It has been reported that *E. cava* extracts may help attenuate obesity, nonalcoholic fatty liver disease, and oxidation stress-mediated diseases [22]. Pyrogallol-phloroglucinol-6,6-bieckol (PPB) is one of the single compounds of *E. cava* extracts and contains several hydroxyl groups. It has been reported that PPB plays a positive role in anti-oxidation by reducing reactive oxygen species (ROS) and improving blood circulation by (i) reducing inflammatory macrophage differentiation, (ii) activating phosphoinositide 3-kinase (PI3K)–protein kinase B (AKT) and 5′ adenosine monophosphate-activated protein kinase (AMPK) pathways in endothelial cells, and (iii) reducing excessive proliferation of vascular smooth muscle cells [23,24]. However, there is a lack of studies characterizing the potential anti-obesity effects of PPB, especially its role in regulating hypertrophic adipocytes mediated by RAGE ligands and RAGE and their ability to control macrophage differentiation. 

In this study, we investigated the anti-obesity effects of PPB (i.e., modification of adipocyte size) mediated by its ability to impact the expression of RAGE ligands and RAGE and the expression of inflammatory cytokines in visceral fat. Since RAGE and inflammation are key factors for adipocyte hypertrophy, the ability of PPB to reduce binding between RAGE and RAGE ligands can decrease the size of visceral fat. In addition, as binding between RAGE and RAGE ligands decreases, macrophage infiltration into the tissue and differentiation into M1-type macrophage also decreases. As a result, pro-inflammatory cytokines are also less abundant in the tissue. Therefore, we expected the ameliorating effects of PPB on adipocyte hypertrophy to be mediated by modulating the expression levels of RAGE ligands, RAGE, and inflammatory cytokines.

## 2. Results and Discussion

### 2.1. Oral Administration of Pyrogallol-Phloroglucinol-6,6-Bieckol Ameliorates Adiposity in a Mouse Model of Diet-Induced Obesity

As a model of DIO, C57BL/6N mice were fed a 45% high-fat diet (HFD) for 8 weeks. After inducing obesity for 4 weeks, the mice were orally administrated saline or PPB (2 mg/kg) depending on their group for another 4 weeks. Animals administered PPB experienced a noticeable decrease in the size of visceral adipocytes and body weight. The body weight of the control was lower than that of the DIO/saline group and the body weight of the DIO/PPB group was statistically lower than the DIO/saline group as well (Figure 1A). Fat-mass data revealed that the mean amount of visceral fat of the control animals was lower than that of animals administered DIO/saline. The mean amount of visceral fat of animals administered DIO/PPB was statistically lower than that of the group receiving DIO/saline. However, the differences between these groups were not statistically significant (Figure 1B). The size of the visceral adipose tissue of control mice was noticeably decreased compared with mice receiving DIO/saline. Additionally, the DIO/saline group had larger mean sizes of visceral adipocytes compared with the control and DIO/PPB groups (Figure 1C,D). A primary strategy of regulating obesity might be to control the size of adipocytes. In a previous study, *E. cava* extracts were shown to regulate the division of adipocytes through regulation of the AMPK pathway and consequently inhibit the accumulation of triglycerides into adipocytes [25,26]. However, there have been no reports to date describing an anti-obesity effect of PPB mediated through RAGE and control of inflammation leading to an increase in the size of adipose tissue. Administration of PPB resulted in a decrease in weight in the DIO animal model, an observation that seemed to be accompanied by a reduction in the size of adipocytes without a concomitant decrease in lean mass. As the consequence of a decrease of visceral fat, triglyceride levels in serum of control and DIO/PPB groups were lower than that of the DIO/saline group (Figure 1E). Additionally, total cholesterol levels of the control and DIO/PPB groups were statistically lower when compared with the DIO/saline group (Figure 1F). PPB appeared to have a positive impact on the metabolism of triglycerides and cholesterol. 

### 2.2. Reduction in the Expression of AGEs, HMGB1, and S100β Following Supplementation with PPB in DIO Mice 

To begin the experiments, the quality of isolated visceral fat protein was validated using Coomassie Brilliant Blue stain and immunoblot assay for β-actin (Appendix A).

RAGE is expressed in many tissues and acts as a receptor for various ligands, including AGEs and other non-glycated proteins (e.g., HMGB1, S100β). RAGE ligands are produced in adipocytes and act as ligands for several receptors in addition to RAGE. In high-fat diet conditions, RAGE ligands are secreted more when compared to lean conditions and bind to RAGE [27]. Adipocyte hypertrophy is known to be induced by a signaling pathway between RAGE and its ligands [28,29]. Downstream signals of RAGE upregulate the transcription of nuclear factor kappa-light-chain-enhancer β (NF-κB) and thus mediate the secretion of inflammatory cytokines [30]. The level of AGEs, HMGB1, and S100β in the visceral fat of mice in the DIO/saline group were remarkably higher compared with the control group. However, mice in the DIO/PPB group had lower levels of AGEs, HMGB1, and S100β compared with the DIO/saline group. (Figure 2A–C). PPB appeared to reduce the production of RAGE ligands in adipose tissue, thus leading to an effective reduction in the secretion of AGEs, HMGB1, and S100β in adipose tissues. Many studies have shown that phlorotannin reduces the amount of RAGE ligands in obesity animal models [31,32]. Although a mechanistic study has been conducted, the precise mechanism of this effect has not yet been revealed. One study, however, suggested that phlorotannin and RAGE inhibition involves the prevention of AGE formation [31]. Phlorotannins extracted from various brown algae have been shown to have similar effects on AGE formation. Phlorotannins were shown to deform AGEs in in vitro tests, an event that led to a decrease in the total amount of RAGE ligands.

### 2.3. Reduction of RAGE Expression and RAGE-RAGE Ligand Bonding Following PPB Supplementation in DIO Mice 

RAGE is well known to play an important role in adipocyte hypertrophy and mediating various inflammatory signals [28]. In concordance with the increased expression of RAGE and RAGE ligands in the fat tissue of obese animals, the ratio of RAGE and RAGE ligand binding also increased. The expression level of RAGE in the DIO/saline group was higher when compared to the control group and the expression level of RAGE in the visceral fat of mice in the DIO/PPB group was statistically lower when compared with the DIO/saline group (Figure 3A). PPB appears to affect the expression of RAGE in visceral fat tissue. The intensity of RAGE expression was similar to these results (Appendix A). The ratio of AGEs, HMGB1, and S100β, and protein binding with RAGE (normalized to control) in visceral fat were lower compared with the DIO/saline group. The ratio of AGEs, HMGB1, and S100β of the DIO/PPB group, and binding to RAGE in visceral fat was lower compared with the DIO/saline group (Figure 3B,C). A reduction in RAGE and RAGE ligand binding following administration of PPB might be thought to be the consequence of decreasing the expression of RAGE and RAGE ligands. However, it is also possible that PPB acts as a RAGE-ligand antagonist. In our study, the level of RAGE ligands in the DIO/PPB group was lower when compared with the DIO/saline group. It appears that PPB may inhibit the formation of AGEs, similar to the mechanism of other phlorotannins. Thus, PPB may regulate the RAGE-signaling pathway. Assessing the nuances of RAGE and PPB binding may require additional studies, including structural analysis of PPB.

### 2.4. Modulation of M1/M2 Macrophage Infiltration Following PPB Supplementation of DIO Mice

As the fat tissue hypertrophied in DIO mice, the number of macrophages infiltrated into visceral fat and the differentiation into M1-type macrophages are increased following increased levels of RAGE ligands secreted by adipocytes [33]. The inflammatory response of adipose tissue is caused by the proportion of macrophage subtypes present in the tissue [34]. The intensity of CD86—an M1-type macrophage marker—in the DIO/saline group was drastically higher compared with the control group. However, the intensity of CD86 in the DIO/PPB group was statistically lower compared with the DIO/saline group (Figure 4A, Appendix A). On the other hand, the intensity of CD163—an M2-type macrophage marker—in controls was higher compared with the DIO/saline group. The intensity of CD163 in the DIO/PPB group was higher compared with the DIO/saline group (Figure 4B, Appendix A). The relative mRNA expression levels of CD86 and CD80 in the control group were statistically lower compared with the DIO/saline group, and the relative mRNA expression level of CD86 and CD80 in the DIO/PPB group was lower compared with the DIO/saline group (Figure 4C). The relative mRNA expression level of CD163 and CD206 in the control group was statistically lower compared with the DIO/saline group. However, the relative mRNA expression level of CD163 and CD206 in the DIO/PPB group was higher compared with the DIO/saline group (Figure 4D). The reduction of RAGE ligands following the administration of PPB reduces the differentiation of macrophages into the M1-type (Figure 4A–C).

### 2.5. Modulation of Inflammatory Cytokine Expression Following PPB Supplementation in DIO Mice

In concordance with the regulation of macrophage differentiation into the M1-type by PPB, the secretion of inflammatory cytokines is decreased in adipose tissue (Figure 5A,B). Inflammatory cytokines secreted from M1-type macrophages induces obesity by upregulating toll-like receptors and disrupting lipid metabolism [35,36,37]. M1-type macrophages secrete inflammatory cytokines (e.g., TNF-α, IL-1β). As a macrophage infiltrates into the fat tissue and the number of M1-type macrophages increases, the expression levels of inflammatory cytokines are also increased in DIO mice [38]. Relative TNF-α and IL-1β mRNA expression levels in the visceral fat of the DIO/saline group were higher compared with the control group. However, relative TNF-α mRNA expression levels in the visceral fat of the DIO/PPB group was lower compared with the DIO/saline group (Figure 5A). 

Similarly, the relative expression levels of IL-1β mRNA in the visceral fat of the DIO/PPB group was lower compared with the DIO/saline group (Figure 5B). Thus, PPB appears to help modulate the macrophage-induced inflammatory response in tissues by regulating the expression of RAGE ligands and RAGE and by reducing their binding, thereby alleviating obesity. Adipocytes, which show inflammatory responses, have increased expression of apoptotic factors (e.g., nuclear factor kappa-light-chain-enhancer β (NF-κB)) and lead to the death of adipocytes [39,40]. Finally, the secretion level of adipokines—typical hormones secreted from adipocytes (e.g., leptin, adiponectin)—is changed and affects various metabolic disorders [41,42,43,44].

## 3. Materials and Methods

### 3.1. Pyrogallol-Phloroglucinol-6,6-Bieckol (PPB) Isolation

Aqua Green Technology Co. Ltd. (Jeju, Korea) provided *E. cava* for this study. The *E. cava* was washed with pure water and air dried for 48 h at room temperature. Dried *E. cava* was ground into a powder and extracted by using 50% ethanol for 12 h and then it was filtered and concentrated in a vacuum machine. Concentrated *E. cava* was sterilized by heating for 60 min at 85 °C and spray dried. To isolate PPB from *E. cava* extract, we used a previously described method [45]. 

In short, the *E. cava* extract was partitioned with ethyl acetate (1:1, *v*/*v* of sample). Then, the dried ethyl acetate fraction was loaded in a centrifugal partition chromatography (CPC). CPC was used with a two-phase solvent system composed of water/ethyl acetate/methanol/n-hexane (7:7:3:2, *v*/*v*/*v*/*v*). For separation, there were two phases—the stationary and mobile phases. The stationary phase (i.e., organic solution) was added to the CPC column while the mobile phase flowed through the column at 2 mL/min rate. The PPB was obtained from fraction II of the *E. cava* extracts and then, PPB solution was lyophilized for 24 h in freeze dryer machine and stored at −20 °C.

### 3.2. Animals

Seven-week-old C57BL/6N male mice were used in this study. To generate the diet-induced obesity (DIO) model, mice were fed a 45% high-fat diet (HFD; Research Diet Inc., New Brunswick, NJ, USA) for 8 weeks. Mice were separated into three groups: control, DIO/saline, DIO/PPB. Control mice were fed normal diet food and water ad libitum. DIO groups continued to receive a 45% HFD after separation. For oral administration, the isolated PPB was dissolved in 0.9% saline (DAI HAN Pharm. Co. LTD., Gyeonggi-do, Korea) and each group was orally administrated saline and PPB (2.5 mg/kg) daily for 4 weeks. Body weights were measured once a week. Fat mass was measured using two-way analysis to determine fat and lean masses separately.

The animal protocol was approved by the animal center of Lee Gil Ya Cancer and Diabetes Institute of Gachon University. All experiments confirmed to the AAALAC international guidelines and veterinary advice. The number of this study is LCDI-2017-0032.

### 3.3. Sample Preparation

#### 3.3.1. RNA Extraction and Complimentary DNA (cDNA) Synthesis 

Visceral fat tissue was homogenized by using the MACS (Miltenyi Biotec, bergisch gladbach, Germany) system and 500 µL of RNisol (TAKARA, Tokyo, Japan). Tissue lysates were added to 100 µl of chloroform and vortexed for 3 s to mix. The lysates were centrifuged at 12,000× *g* for 15 min at 4 °C. Supernatants were then collected and transferred to 1.7 mL tubes. Supernatants were mixed with 0.5 mL of isopropanol and centrifuged at 12,000× *g* for 15 min at 4 °C. The resulting RNA pellet was washed with 70% ethanol and centrifuged at 12,000× *g* for 5 min at 4 °C. The RNA pellet was then dissolved in 30–50 µL of Diethyl pyrocarbonate-treated water based on the size of pellet (i.e., larger pellet dissolved in greater volume). cDNA was synthesized from 1 µg of total RNA using a Prime Script 1st strand cDNA Synthesis Kit (TAKARA).

#### 3.3.2. Protein Isolation

Visceral fat proteins were extracted using the EzRIPA lysis kit (ATTO, Tokyo, Japan). Tissues were homogenized by using the MACS (Miltenyi Biotec) system and a lysis buffer containing proteinase and phosphatase inhibitors. Lysates were then sonicated for 10 s in an ice bath sonication machine followed by centrifuging at 14,000× *g* for 20 min at 4 °C. Supernatants were then collected and protein concentrations were measured using a bicinchoninic acid assay kit (Thermo Fisher Scientific, Waltham, MA, USA). 

#### 3.3.3. Paraffin Slide Preparation

Visceral fat tissue was preserved in 4% para-formaldehyde solution for 1 week for fixation. After running water to wash the tissue for 30 min, the tissue processor machine (Shandon Citadel, Ramsey, MN, USA) was operated for 14 h. Next, the tissue was embedded with paraffin and stored at room temperature. The tissue was cut into 10 µm sections and placed on coated slides. The slides were incubated at 37 °C with a heat plate overnight. 

### 3.4. Enzyme-Linked Immunosorbent Assay (ELISA)

#### 3.4.1. Indirect ELISA Assay

To measure the secretion of RAGE ligands from visceral fat, 96-well microplates were coated with anti-AGEs, anti-HMGB1, and anti-S100β antibodies diluted in 100 nM carbonate and bicarbonate mixed buffer, adjusted to pH 9.6, and incubated overnight at 4 °C. Microplates were then washed with phosphate buffer saline (PBS) containing 0.1% Triton x-100 (TPBS). The remaining protein-binding sites were then blocked by using 5% skim milk for 6 h at room temperature. After washing with PBS, protein samples of the control, saline, and PPB were distributed to each well and incubated overnight at 4 °C. Each well was rinsed with TPBS then incubated for 4 h at room temperature with a peroxidase-conjugated secondary antibody. Tetramethylbenzidine (TMB) solution was added followed by incubation for 15–20 min at room temperature. 2N H_2_SO_4_ was used as a stop solution. Optical density was validated using a microplate reader at 450 nm wavelength (Spectra max plus, Molecular Devices, San Jose, CA, USA).

#### 3.4.2. Sandwich Enzyme-Linked Immunosorbent Assay

To analyze RAGE ligand expression and the interactions between RAGE and RAGE ligands, the wells of a 96-well microplate were coated with antibodies (i.e., anti-AGE, anti-HMGB1, anti-S100β antibodies) in 100 mM carbonate/bicarbonate buffer (pH 9.6) overnight at 4 °C. After washing wells with TPBS, 5% skim milk was added to block the remaining protein-binding sites for 6 h at room temperature. Samples were added to each well and incubated overnight at 4 °C. After washing with PBS, an anti-RAGE antibody (dilution rate 1: 200) was added and left overnight at 4 °C. After washing with TPBS, samples were incubated for 4 h at room temperature with peroxidase-conjugated secondary antibody (dilution rate 1:1000). Substrate TMB solution was added to each well and incubated for 15–20 min. An equal volume of stop solution (2N H_2_SO_4_) was added. Optical densities were measured at 450 nm. Antibody concentrations are listed in Appendix A.

### 3.5. Immunoblotting

Visceral fat protein (25 µg) was loaded to 8% SDS-PAGE gel then transferred into the polyvinylidene difluoride membrane (Millipore) using semi-dry transfer machine (ATTO). After the transfer process, the membrane was incubated in 5% of skim milk for 1 h at room temperature. The membrane was washed with tris buffer saline containing 0.1% Triton x-100 (TTBS) for 10 min 3 times. Then the membrane was incubated with anti β-actin antibody (dilution rate 1: 1000) overnight at 4 °C. The membrane was incubated with horseradish peroxidase (HRP) goat anti-rabbit IgG antibody (Peroxidase, dilution rate 1:5000) for 2 h at room temperature and then followed by rinsing with TTBS for 10 min 3 times. The membrane was developed using LAS—4000 (GE Life Science, Chicago, IL, USA). The protein marker (TransGen Biotech Co., LTD, Beijing, China) was used to check the molecular weight of proteins.

### 3.6. Quantitative Real Time Polymerase Chain Reaction

mRNA was isolated from visceral fat by manual protocol. Next, cDNA was synthesized using a Prime 1st strand cDNA synthesis kit (TAKARA). CFX386 touch (Bio-Rad, Hercules, CA, USA) was used to conduct qRT-PCR with primers (Appendix A). Reaction efficiency and the number of cycles were determined by innate software.

### 3.7. Immunofluorescence

Paraffin-embedded tissue slides were heated in the microwave with antigen-retrieval solution for 5 min. RAGE expression levels were validated in visceral fat. Paraffin-sectioned slides were deparaffinized by treatment with xylene prior to the addition of an antigen-retrieval solution. The slides were washed with PBS then placed into a blocking solution for 1 h at room temperature. After rinsing, slides were incubated with anti-RAGE (Santa Cruz Biotechnology Inc) primary antibody (dilution rate 1:100) overnight. Slides were then washed with PBS and incubated with 488 fluorescence anti-mouse secondary antibody (Abcam, San Francisco, CA, USA) (dilution rate 1:200) for 2 h at room temperature. Next, the slides were rinsed again with PBS. Nuclei were stained with 4′,6-diamidino-2-phenylindole (dilution rate 1:1000) for 3 min at room temperature. Finally, slides were mounted with vectashield (Vector Laboratories Inc, Burlingame, CA, USA).

### 3.8. Histology

Hematoxylin and eosin staining was conducted as recommended for visceral fat tissue. Slides were deparaffinized by xylene, dipped in Hematoxylin (DAKO, Carpinteria, CA, USA) solution for 2 min prior to eosin for 5 s and finally mounted by using the DPX solution (Sigma-Aldrich, St. Louis, MO, USA).

### 3.9. 3,3-Diaminobenzidine (DAB) Staining

Paraffin-embedded tissue slides were heated in a microwave with antigen-retrieval solution for 5 min. CD86 and CD163 were validated to measure the expression level in visceral fat. Paraffin-sectioned slides were deparaffinized by xylene prior to using antigen-retrieval solution. The slides were washed with PBS solution then placed into a 0.3% H_2_O_2_ solution for 30 min. Next, the slides were washed with PBS solution and added to the blocking solution for 1 h at room temperature. After rinsing, slides were incubated with each antibody overnight. After washing again with PBS solution, the Avidin-Biotin Complex Staining Kits (ABC Kits; Vector laboratories) was used for 30 min as per the manufacturer’s protocol to amplify the signal (dilution rate 1:50). Slides were then washed with PBS solution and developed with the DAB solution for 4 min. Slides were then washed again with running water and then subjected to hematoxylin staining for 15 s. After a final wash with running water, slides were mounted with DPX solution (Sigma-Aldrich). Images were collected using light microscopy (LSM 710, Carl Zeiss, Oberkochen, Germany).

### 3.10. Coomassie Blue Staining

To validate the purification of visceral fat protein, Coomassie blue staining was implemented. Visceral fat protein (75 µg) was loaded to 8% SDS-PAGE gel and then the gel was incubated in fixation solution (acetate/methanol/water, 1:4.5:4.5, *v*/*v*/*v*) for 10 min at room temperature. The gel was incubated with 1% Commassie Brilliant Blue (Amresco, Zottegem, Belgium) staining solution (acetate/methanol/water/Coomassie blue staining solution, 1:4:4:1, *v*/*v*/*v*/*v*) for 4 h at room temperature. The gel was washed with destaining solution (acetate/metanol/water, 1:4:4) overnight at room temperature.

### 3.11. Fat Size and Intensity Measurement

To measure the size of visceral fat, several photos of random sites from hematoxylin- and eosin-stained slides were taken by microscopy (200× magnification) followed by measurement of size using Image J software (NIH, Bethesda, MD, USA). Intensity was also measured by innate software in Image J.

### 3.12. Triglyceride Measure Assay

To measure triglyceride levels in serum, we obtained blood in an ethylenediaminetetraacetic acid (EDTA) buffer-coated tube (300–500 μL). Tubes were centrifuged at 2000 rpm for 20 min at room temperature. After collection of the supernatant from the tube, total triglyceride levels were measured by KPNT company (Cheongju, Gyeonggi-do, Korea) following the peroxidase-coupled method [46]. Optical density was validated by microplate reader at 510 nm wavelength (Molecular devices).

### 3.13. Total Cholesterol Measure Assay

To measure the total cholesterol level in serum, we obtained blood in an EDTA buffer-coated tube (300–500 μL). Tubes were centrifuged at 2000 rpm for 20 min at room temperature. After collection of the supernatant from the tube, total cholesterol levels were measured by KPNT (Cheongju, Gyeonggi-do, Korea) following the cholesterol oxidase / peroxidase (CHOD-POD) method [47]. Optical density was validated by microplate reader at 510 nm wavelength (Molecular devices).

### 3.14. Statistical Analysis

The Kruskal–Wallis and Mann–Whitney U post-hoc tests were used to compare results across groups. The SPSS version 22 (IBM Corporation, Armonk, NY, USA) program was used to determine statistically significant differences between groups. Results are expressed as the mean ± standard deviation (SD). Differences were considered significant at * *p* < 0.05, ** *p* < 0.01, and *** *p* < 0.001 vs. control; ^#^
*p* < 0.05, ^##^
*p* < 0.01, and ^###^
*p* < 0.001 vs. DIO/saline.

## 4. Conclusions

PPB single compound contained in *E. cava* exerted anti-obesity effects by reducing the expression of RAGE and the secretion of its ligands. The expression of RAGE is known to be critical in visceral fat hypertrophy. PPB can reduce adipocyte hypertrophy and inflammation in fat tissue by controlling the differentiation of macrophages via regulation of the expression of RAGE and RAGE ligands and their cognate binding in visceral fat. PPB may serve as an antagonist of RAGE ligands, as evidenced by a decrease in the binding of RAGE to its ligands. In addition, PPB can reduce the number of activated macrophages and inflammatory cytokine levels. Additional studies are needed to further confirm the observations presented here. However, these results suggest that PPB, as a functional ingredient, may alleviate obesity. 

## Figures and Tables

**Figure 1 marinedrugs-17-00612-f001:**
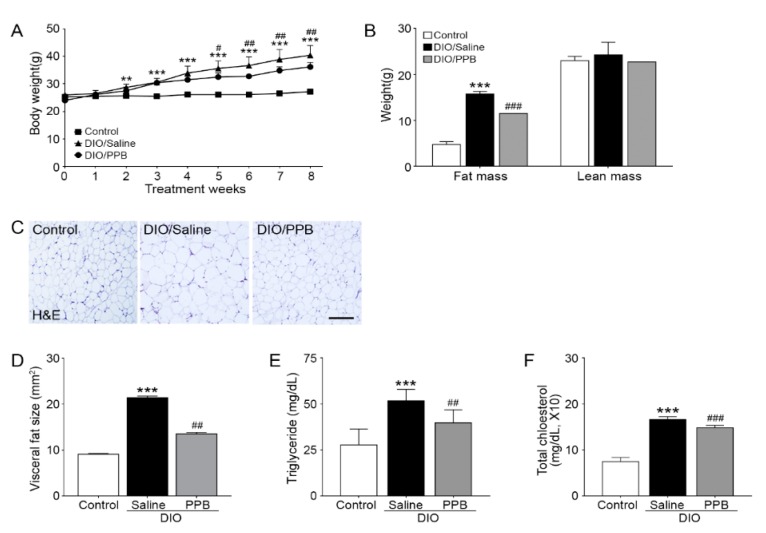
Reducing effects of pyrogallol-phloroglucinol-6,6-bieckol (PPB) supplement on visceral fat size and body weight in diet-induced obesity (DIO) mice. (**A**) Body weights were measured daily for 8 weeks. (**B**) Graphs indicate the fat and lean masses. (**C**) The representative images are hematoxylin and eosin stained visceral fat. (**D**) Size of visceral fat tissue. (**E**) Triglyceride levels in serum. (**F**) Total cholesterol level in serum. Scale bar = 100 μm; 200× magnification. Significance represented as ** *p* < 0.01 versus control; *** *p* < 0.001 versus control; ^#^
*p* < 0.05 versus DIO/saline; ^##^
*p* < 0.01 versus DIO/saline; ^###^
*p* < 0.001 versus DIO/saline. DIO, diet-induced obesity; H&E, hematoxylin & eosin; PPB, pyrogallol-phloroglucinol-6,6-bieckol.

**Figure 2 marinedrugs-17-00612-f002:**
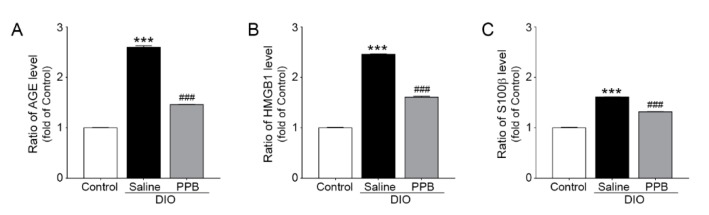
Inhibiting effects of PPB supplement on AGEs, HMGB1 and S100β expression in visceral fat of the DIO animal model. Relative (**A**) AGE, (**B**) HMGB1, and (**C**) S100β protein levels in visceral fat normalized to the control group. The significance represented as *** *p* < 0.001 versus Control; ^###^
*p* < 0.001 versus DIO/saline. AGEs, advanced glycation end-products; DIO, diet-induced obesity; HMGB1, high mobility group box1; PPB, pyrogallol-phloroglucinol-6,6-bieckol; S100β, S100 beta.

**Figure 3 marinedrugs-17-00612-f003:**
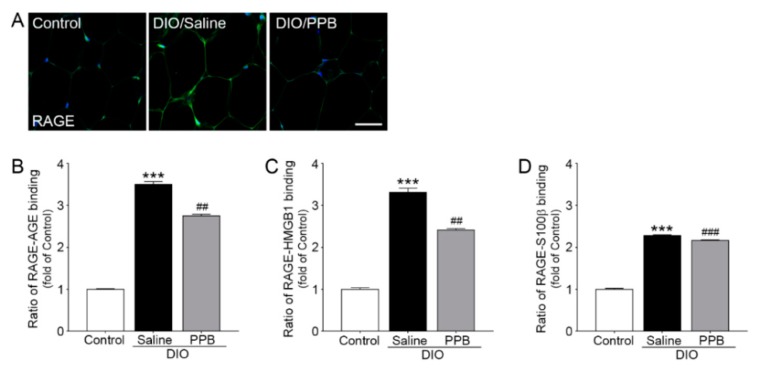
Inhibiting effects of PPB supplementation on RAGE expression and binding to its RAGE ligands in visceral fat of DIO mice. (**A**) Representative examples of immunofluorescence staining (and intensity) of RAGE; RAGE-positive markers (green) and DAPI stained nuclei (blue). (**B**) RAGE-AGEs, (**C**) RAGE-HMGB1, and (**D**) RAGE-s100β binding in visceral fat normalized to control. Scale bar = 100 μm; 200× magnification. The significance represented as *** *p* < 0.001 versus control; ^##^
*p* < 0.01 versus DIO/saline, ^###^
*p* < 0.001 versus DIO/saline. AGEs, advanced glycation end-products; DAPI, 4′,6-Diamidine-2′-phenylindole dihydrochloride; DIO, diet-induced obesity; HMGB1, high mobility group box1; PPB, pyrogallol-phloroglucinol-6,6-bieckol, RAGE, receptor for advanced glycation end-products; S100β, S100 beta.

**Figure 4 marinedrugs-17-00612-f004:**
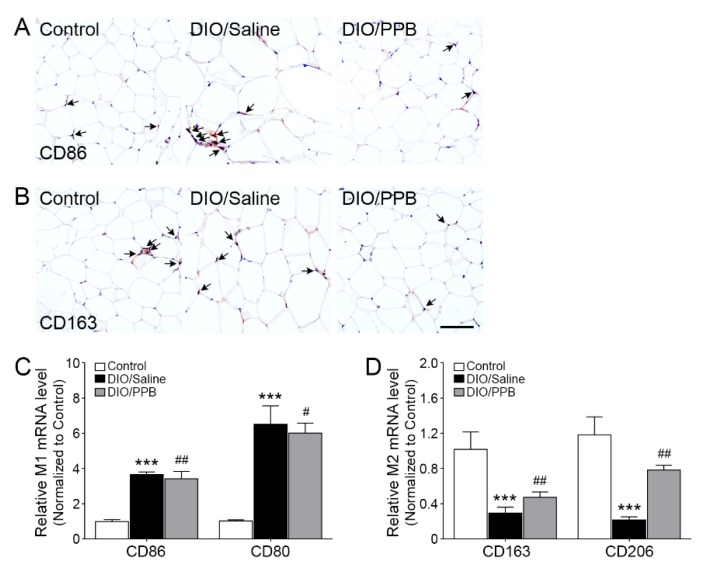
Modulating effects of PPB supplementation on macrophage differentiation in visceral fat of DIO mice. (**A**,**B**) DAB staining of (**A**) CD86—an M1-type marker—and (**B**) CD163—an M2-type marker. (**C**,**D**) Relative mRNA expression level of (**C**) CD86 and CD80 and (**D**) CD163 and CD206 in visceral fat. Scale bar = 100 μm; 200× magnification. The significance represented as *** *p* < 0.001 versus control; ^#^
*p* < 0.05 and ^##^
*p* < 0.01 versus DIO/saline. DAB, 3, 3-diaminobenzidine; DIO, diet-induced obesity; PPB, pyrogallol-phloroglucinol-6,6-bieckol.

**Figure 5 marinedrugs-17-00612-f005:**
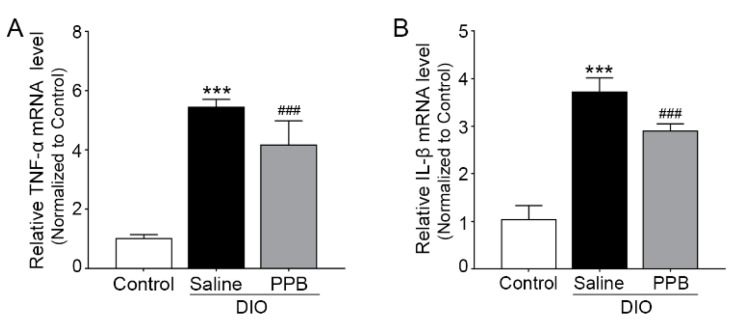
Modulating effects of PPB supplement on inflammatory cytokine expression in visceral fat of DIO mice. (**A**) Relative mRNA expression levels of TNF-α and (**B**) IL-1β in visceral fat. *** *p* < 0.001 versus Control; ^###^
*p* < 0.001 versus DIO/saline. The graph indicates relative mRNA expression level of CD163 in visceral fat. Scale bar = 100 μm, the significance represented as *** *p* < 0.001 versus control; ^###^
*p* < 0.01 versus DIO/saline. DIO, diet-induced obesity; IL-1β, interleukin-1 beta; PPB, pyrogallol-phloroglucinol-6,6-bieckol; TNF-α, tumor necrosis factor-alpha.

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
