# Peer review of "Pyrogallol-Phloroglucinol-6,6-Bieckol Alleviates Obesity and Systemic Inflammation in a Mouse Model by Reducing Expression of RAGE and RAGE Ligands"

_marinedrugs, 2019, doi:10.3390/md17110612_

Round 1

Reviewer 1 Report

Find attached the edited manuscript.

Author Response

Response to Reviewer 1 Comments

Point 1 : Do you mean to say "Part of this increasing interest stems from.....

Response 1 : We appreciate your comment. As you commented, the sentence that you gave comment us is similar to following sentence. We wrote two similar sentence into one to make it clear. You are able to check the sentence (Introduction section, Line 63).

Point 2 : to "be"

Response 2 : We appreciate your comment. As you point out, we added ‘be’ between ‘to’ and ‘a’. ‘E.cava is commonly found near the coast of JeJu island and is known to a particularly rich source of phlorotannin’ was changed to ‘E.cava is commonly found near the coast of JeJu island and is known to be a particularly rich source of phlorotannin’. You can check correct sentence (Introduction section, Line 67).

Point 3 : in solution or as pellets?

Response 3 : We appreciate your comment. PPB was used in solution form to oral administration. In a nut, PPB isolated from the E. cava extract by using centrifugal partition chromatography (CPC) method was in freeze dry to make powder. Then PPB was solved in 0.9% saline (DAI HAN Pharm. Co. LTD., catalogue number. A4S3P52) and orally administrated to each animal. You can check the correct sentence in manuscript (Materials and method section, Line 224).

Point 4 : **, ## are not in the figure

Response 4 : We appreciate your comment. We removed those statistical marks. You can check the correct sentence in manuscript (Figure legend section, Line 142).

Point 5 : The differentiation in these stainings is not clear. The originals of these should be presented as a supplement.

Response 5 : We appreciate your comment. As you comment, it is hard to define the differentiation with staining images. So, we added M1 (CD86) and M2 (CD163) type of macrophage markers stained images and quantified graphs in supplementary figure 2 and 3. The expression level of CD86 and CD163 is similar pattern to existed staining results. As well as stained images, we validated with another M1 (CD80) and M2 (CD206) type of macrophage markers by using quantitative polymerase chain reaction (qRT-PCR). CD80 well known to M1 type of macrophage marker and CD206 as M2 type of macrophage marker show similar pattern to staining results. We added the results as a Figure 4C and D and you can check the revised sentence in manuscript (Results and discussion section, figure legend and supplementary table 2 and figure 2, Line 175, 193).

Point 6 : incomplete sentence!

Response 6 : We appreciate your comment. As you commented, we corrected imperfect sentence. You can check in manuscript (Result and discussion section, Line 205).

Point 7 : Reference 46 cannot found

Response 7 : We appreciate your comment. As you commented there was no reference 46 in the manuscript. We added the reference paper which describes isolation method of PPB from E. cava extracts. The reference 46 information added in Reference section of manuscript (Reference section, Line 498).

Point 8 : At what stage was the PPB obtained? Your references ended at 45 so there is not ref 46 to look up.

Response 8 : We appreciate your comment. We used PPB obtained fraction Ⅱ of E. cava extract by using centrifugal partition chromatography (CPC) system and added this information in Materials and methods section of manuscript (Materials and methods section, Line 225). In additional, the reference paper 46 is about that PPB isolation methods from E. cava. extract [Ref: Food Chem 2014, 158, 433- 437] and we added reference paper information in Reference section (Reference section, Line 498).

Reviewer 2 Report

In this manuscript, the authors investigated the anti-obesity effects of PPB mediated by its ability to impact the expression of RAGE and RAGE ligands, as well as the expression of inflammatory cytokines in visceral fat. The findings of this paper are important for the disease of obesity.

Overall, the paper is written well. Nevertheless, my main concern about the purification of protein extracted from Visceral fat. I would suggest to the authors to justify the purification of proteins using SDS-PAGE. An experiment is required to resolve this technical issue. 

Author Response

Response to Reviewer 2 Comments

Point 1 : Overall, the paper is written well. Nevertheless, my main concern about the purification of protein extracted from Visceral fat. I would suggest to the authors to justify the purification of proteins using SDS-PAGE. An experiment is required to resolve this technical issue.

Response 1: We appreciated this comment. We validated the purification of visceral fat protein using Coomassie Brilliant Blue stained SDS-PAGE (A) and immunoblotting for β-actin (B). As you can see (below image), there are many protein bands of samples (N=2) used visceral fat tissues in Coomassie Brilliant Blue stained SDS-PAGE (A) and β-actin band is clear in all samples (B). It means that purification of protein extracted from visceral fat tissues is fine for study.

Round 2

Reviewer 2 Report

According to my comments, the authors validated the purification of extracted proteins by SDS-PAGE. This result should be included in the manuscript. I would suggest to the authors to mentioned this result clearly in the main text (including method) and provide the image (including figure legend) in the supplementary info. 

What protein marker was used for this analysis? The authors should mention the protein marker, including the manufacturer. 

Author Response

Point 1 :According to my comments, the authors validated the purification of extracted proteins by SDS-PAGE. This result should be included in the manuscript. I would suggest to the authors to mentioned this result clearly in the main text (including method) and provide the image (including figure legend) in the supplementary info.

Response 1 : We appreciate your comment. As you commented, we mentioned the result and method in the main text (Results and Discussion section, Line 124 and Materials and method section, Line 303, 346) and added images as figure S3 with the legend.

Point 2 :What protein marker was used for this analysis? The authors should mention the protein marker, including the manufacturer.

Response 2: We appreciate your comments. We used EasySee® western marker (TransGen Biotech Co., LTD, Cat # DM201, Beijing, China). You can check the information in data sheet and we added this information in materials and methods section, Line 311.
